# DNA Methylation Signatures of Breastfeeding in Buccal Cells Collected in Mid-Childhood

**DOI:** 10.3390/nu11112804

**Published:** 2019-11-17

**Authors:** Veronika V. Odintsova, Fiona A. Hagenbeek, Matthew Suderman, Doretta Caramaschi, Catharina E. M. van Beijsterveldt, Noah A. Kallsen, Erik A. Ehli, Gareth E. Davies, Gennady T. Sukhikh, Vassilios Fanos, Caroline Relton, Meike Bartels, Dorret I. Boomsma, Jenny van Dongen

**Affiliations:** 1Department of Biological Psychology, Vrije Universiteit Amsterdam, 1081 BT Amsterdam, The Netherlandsdi.boomsma@vu.nl (D.I.B.); 2Amsterdam Public Health Research Institute, 1081 BT Amsterdam, The Netherlands; 3Kulakov National Medical Research Center for Obstetrics, Gynecology and Perinatology, Moscow 101000, Russia; 4MRC Integrative Epidemiology Unit, Bristol Medical School, Population Health Science, University of Bristol, Bristol BS8 1TH, UK; 5Avera Institute for Human Genetics, Sioux Falls, SD 57101, USA; 6Neonatal Intensive Care Unit, Department of Surgical Sciences, AOU and University of Cagliari, 09121 Cagliari, Italy

**Keywords:** breastfeeding, EWAS, DNA methylation, twins, EPIC, NTR, ALSPAC

## Abstract

Breastfeeding has long-term benefits for children that may be mediated via the epigenome. This pathway has been hypothesized, but the number of empirical studies in humans is small and mostly done by using peripheral blood as the DNA source. We performed an epigenome-wide association study (EWAS) in buccal cells collected around age nine (mean = 9.5) from 1006 twins recruited by the Netherlands Twin Register (NTR). An age-stratified analysis examined if effects attenuate with age (median split at 10 years; *n*_<10_ = 517, mean age = 7.9; *n*_>10_ = 489, mean age = 11.2). We performed replication analyses in two independent cohorts from the NTR (buccal cells) and the Avon Longitudinal Study of Parents and Children (ALSPAC) (peripheral blood), and we tested loci previously associated with breastfeeding in epigenetic studies. Genome-wide DNA methylation was assessed with the Illumina Infinium MethylationEPIC BeadChip (Illumina, San Diego, CA, USA) in the NTR and with the HumanMethylation450 Bead Chip in the ALSPAC. The duration of breastfeeding was dichotomized (‘never‘ vs. ‘ever’). In the total sample, no robustly associated epigenome-wide significant CpGs were identified (α = 6.34 × 10^–8^). In the sub-group of children younger than 10 years, four significant CpGs were associated with breastfeeding after adjusting for child and maternal characteristics. In children older than 10 years, methylation differences at these CpGs were smaller and non-significant. The findings did not replicate in the NTR sample (*n* = 98; mean age = 7.5 years), and no nearby sites were associated with breastfeeding in the ALSPAC study (*n* = 938; mean age = 7.4). Of the CpG sites previously reported in the literature, three were associated with breastfeeding in children younger than 10 years, thus showing that these CpGs are associated with breastfeeding in buccal and blood cells. Our study is the first to show that breastfeeding is associated with epigenetic variation in buccal cells in children. Further studies are needed to investigate if methylation differences at these loci are caused by breastfeeding or by other unmeasured confounders, as well as what mechanism drives changes in associations with age.

## 1. Introduction

Early life environmental influences are associated with development and disease. One mechanism that has been hypothesized to account for the association is through the “epigenetic programming” of genes, some of which may act to confer plasticity to developmental processes [1,2]. Nutrition in early life may play a crucial role in modulating gene expression [3,4]. It has been hypothesized that nutrition-induced epigenetic variation may result in different development trajectories and may be associated with metabolic and immune development during critical periods of early life [5,6,7,8]. Epigenetic mechanisms are a key element in understanding the developmental origins of later life disease risk [9,10,11]. One of the best studied epigenetic mechanisms is DNA methylation—the modification of a cytosine base, usually at CpG dinucleotides, with a methyl group, which regulates gene expression and seems to already be sensitive to nutrition during the prenatal period [12,13,14,15,16,17].

In humans, prenatal maternal famine has been associated with long-term DNA methylation changes that are still observable in middle-aged individuals [18,19]. The early postnatal period is also believed to be a critical period at which permanent long-term changes may be induced by environmental exposures that affect the child’s susceptibility to chronic disease [20]; however, whether similar long-term changes in the human epigenome may be induced in this period is unexplored. The first nutrition period, including breastfeeding, has long-term effects on children [3]. Human breast milk has a unique composition that differs from other lactating animals and that is quite impossible to reproduce in artificial production; in fact, breast milk contains a unique mixture of microorganisms (the microbiome) [21], metabolites [22], multipotent stem cells [23], growth factors [24] and other components that render it unique and individualized for each newborn [25]. Moreover, breast milk varies its composition according to the lactation period and circadian rhythm, and it even varies from the start till the end of one feeding [26]. The benefits of breastfeeding on the health of children are widely described [27,28,29] and may involve the transmission of nutrients, hormones, and antibodies from mother to child [24,30,31]. These benefits also include the process of interaction and attachment of a child to their mother, although the effect of bonding may also be achieved through formula feeding [32,33].

Balanced newborn feeding is the basis for the adequate growth and development in childhood and beyond [25,28,34]. Breast milk is important for sensory, neurological and cognitive development [35,36,37], especially in preterm infants [38], but its effects on cognition are confounded by maternal education [39]. The association between breastfeeding duration and a lower risk of infectious diseases, obesity, cancer, coronary heart disease, some allergies, autoimmune disease, diabetes mellitus, inflammatory bowel diseases, and metabolic syndrome at later age has been widely studied [27,40,41,42,43,44,45,46,47,48,49]. Even though a protective effect of breastfeeding in hypertension, diabetes, obesity, and metabolic syndrome has not been evident in some large studies [30,50], the benefits of breastfeeding are well recognized, and the understanding of the biological mechanisms of its influence is of interest. Previous studies on the association between breastfeeding and child outcomes have used a variety of definitions of breastfeeding, including: never vs. ever breastfeeding, breastfeeding duration which has been assessed as a continuous measure in months or weeks, or as a categorical variable (e.g., long vs. short). Some studies have examined the percentage of breastfed meals, exclusive breastfeeding duration (when a child is exclusively breastfed until formula feed and/or solid foods are introduced). In a meta-analysis by Victora et al. [27], breastfeeding never vs. ever was associated with a reduction in sudden infant deaths, a reduction of acute otitis during the first few years of life, a protection against allergic rhinitis in children <5 years, a higher child Intelligence Quotient (IQ) of about three IQ points. More versus less breastfeeding has been associated with major protection against diarrhea morbidity, a reduction in severe respiratory infection, and effects on deciduous teeth. Exclusive breastfeeding has been associated with a strong protective result against infectious disease and allergic rhinitis in children <5 years. A dose–response association with duration of breastfeeding was found for a higher IQ and a decreased risk for overweight and obesity.

Epigenome-wide association studies (EWASs) can offer new insights into DNA–environment interactions in determining child development and health [51]. To our knowledge, seven association studies (two candidate gene studies, two EWASs with breastfeeding as covariate and three EWASs of breastfeeding) have been performed on breastfeeding and DNA methylation in humans to date (see Appendix A). A candidate gene study of Obermann-Borst et al. found a negative association between the duration of breastfeeding (in categories) and methylation level in blood cells from 120 children (mean age 1.4 years) at seven CpG sites in the promoter of the *LEP* gene; a hormone that regulates energy homeostasis [52]. A suggestive positive association of the methylation level of 2201 CpGs and a negative association of the methylation level of 2075 CpGs with the duration of breastfeeding (continuous measure in weeks) were reported in blood samples from 37 infants (mean age 25.7 months) [53]. These CpGs were annotated to genes predominantly involved in the control of cell signaling systems, the development of anatomical structures and cells, and the development and function of the immune and central nervous systems [53]. The impact of breastfeeding duration (continuous measure in months) on DNA methylation patterns in 200 children (mean age 11.6 years) was suggested in a study of asthma [54]. An EWAS of Sherwood et al. on exclusive breastfeeding supported the findings of Obermann-Borst et al. at a later stage in childhood (10 years, *n* = 297) but not in young adulthood (18 years, *n* = 305) [55]. This suggests that methylation changes induced by breastfeeding may change with time and may be more evident at an early age. Similarly, it has been observed that associations between DNA methylation and maternal smoking and birthweight attenuate during childhood [56,57]. Nevertheless, a long-lasting modulating effect of breastfeeding (continuous measure in weeks) on the effects of methylation quantitative trait loci (mQTLs) for CpG sites at the 17q21 locus, where the *IL4R* (interleukin-4) gene is located, has been suggested at age 18 (*n* = 245) [58]. Not having been breastfed has been associated with an increase in methylation of the promoter of the tumor suppressor gene *CDKN2A* (cyclin dependent kinase inhibitor 2A) in premenopausal breast tumors of 639 women (mean age of 57.6 years) [59]. In a more recent EWAS study, breastfeeding (dichotomized as never vs. ever) was associated with changes in the *TTC34* gene at age 7 (*n* = 640), which were still evident in adolescence (*n* = 709) [60]. These previous epigenetic studies of breastfeeding were often conducted with relatively small samples (average sample size = 307, range = 37–640). In all studies, DNA was extracted from peripheral blood [52,53,54,55,58], or from tumor tissues in adults [59].

We aimed to carry out an EWAS of breastfeeding in 1006 children around nine years of age recruited by the Netherlands Twin Register (NTR) based on buccal cell DNA and a replication analysis of loci previously associated with breastfeeding in aforementioned epigenetic studies. Buccal samples typically consist of a large proportion of epithelial cells, which might serve as a surrogate tissue for other ectodermal tissues, including the brain [61,62]. Buccal samples also consist of a smaller proportion of leukocytes [63]. To date, few EWASs have been performed on buccal DNA. As some studies have suggested that the effects of early life exposures, including breastfeeding [55,56,57,58], may fade away during childhood, we also performed an EWAS on younger children (age <10 years; where 10 corresponds to the median age of the sample) and compared effect sizes in this group with effect sizes in children older than 10 years. We applied a median split of the sample by age to achieve equal sample sizes in both groups. We hypothesized that if effects of breastfeeding attenuate with age, associations would be strongest in the younger age group. We performed replication in an independent buccal-cells DNA methylation dataset from the NTR (*n* = 98) and in a blood-DNA-methylation dataset from the Avon Longitudinal Study of Parents and Children (ALSPAC) (*n* = 938). We also examined the correlation between methylation levels of twins for the significant CpGs associated with breastfeeding. We hypothesized that the equal exposure to breastfeeding of co-twins should cause resemblance in their methylation profiles.

## 2. Materials and Methods

### 2.1. Overview

We carried out an EWAS in the NTR in 1006 children from 496 complete pairs and 14 twins from incomplete pairs with DNA methylation in buccal cells, testing 787,711 methylation sites (α = 0.05/787771 = 6.34 × 10^–8^). The EWAS analyses in different age groups were performed following the median-split of the sample by age (age <10 years: *n* = 517, age range = 5–9; age ≥ 10 years: *n* = 489, age range = 10–12). Two models were applied with different covariates (Model 1 = “basic model” and Model 2 = “adjusted model”) (see Appendix A). Epigenome-wide significant results were taken forward, after checking effects of outliers, for a replication analysis in an independent sample, consisting of 98 NTR children with DNA methylation in buccal cells and in 938 ALSPAC children with DNA methylation in peripheral blood cells. Lastly, we performed a follow-up analysis of the results of previous studies (3859 CpGs) in the discovery cohort (NTR) in the total sample (*n* = 1006) and the younger age group (*n* = 517). A flowchart of the analyses is provided in Figure 1.

### 2.2. Subjects and Samples

#### 2.2.1. Discovery Study

The subjects were enrolled in the NTR [64] a few weeks to months after birth. Informed consent was obtained from parents. Data on breastfeeding collected around the age of 2 years of the children and good quality DNA methylation data around 9 years of age (mean = 9.5, standard deviation (SD) = 1.89, range = 5–12) were available for 1006 children. The dataset included 51.6% girls and 86% monozygotic twins. This study is embedded in a larger project on childhood aggression: Aggression in Children: Unraveling gene-environment interplay to inform Treatment and InterventiON strategies (ACTION) [65,66]. From the population-based NTR, the ACTION study identified twins who at least once scored higher or lower on a sum score for aggression [67,68]. 

#### 2.2.2. NTR Replication Study

An independent group of children from the NTR for whom EPIC array methylation data were previously described [63] were included as replication study. This cohort was also embedded in the ACTION project and were comprised of 98 monozygotic twins with available data on breastfeeding duration and DNA methylation from buccal swabs (mean age = 7.5 years, SD = 2.4, age range = 1–10). The NTR studies were approved by the Central Ethics Committee on Research Involving Human Subjects of the VU University Medical Centre, Amsterdam, an Institutional Review Board certified by the U.S. Office of Human Research Protections (IRB number IRB00002991 under federal-wide Assurance-FWA00017598; IRB/institute codes, NTR 03–180).

#### 2.2.3. ALSPAC Replication Study

Data came from the ALSPAC [69,70], a population-based birth cohort. All pregnant women living in the geographical area of Avon (UK) with expected delivery date between 1 April 1991 and 31 December 1992 were invited to participate. Approximately 85% of the eligible population was enrolled, totaling 14,541 pregnant women who gave informed and written consent. The study website contains details of all the data that are available through a fully searchable data dictionary and variable search tool (http://www.bris.ac.uk/alspac/researchers/data-access/data-dictionary/). Ethical approval for the study was obtained from the ALSPAC Ethics and Law Committee and the Local Research Ethics Committees. Of these, 938 children had information on breastfeeding duration and DNA methylation from peripheral blood cells measured on the Infinium HumanMethylation450 BeadChip (mean age = 7.4, SD = 0.13, age range = 7.1–8.8) within the Accessible Resource for Integrated Epigenomics Studies (ARIES) project [71].

### 2.3. Phenotype Data

In the NTR, breastfeeding was assessed in a questionnaire sent to mothers 2 years after the twins were born. Mothers were separately asked about breastfeeding for the first and second born twin. There were six answer categories: ‘no’, ‘less than 2 weeks’, ‘2–6 weeks’, ‘6 weeks to 3 months’, ‘3–6 months’, and ‘more than 6 months’, For the main analyses, breastfeeding as exposure was recoded into 2 categories: ‘ever’ and ‘never’, No information on exclusive and mixed breastfeeding was available. The duration of breastfeeding was used in a secondary analysis based on the 6 categories from the original questionnaire coded from 0 (no) to 5 (more than 6 months). Socio-economic status (SES) was determined in two ways (depending on the version of the survey): (1) SES was obtained from a full description of the occupation of the parents and was subsequently coded according to the Standard Classification of Occupations [72]; (2) SES was obtained by the Erikson-Goldthorpe-Portocarero (EPG)-classification scheme [73], combined with information on parental education. The self-reported maternal pre-pregnancy weight just before pregnancy (in kilograms) divided by the square of height (in meters) was used to obtain the maternal pre-pregnancy body mass index (BMI) (weight/height^2^). Maternal smoking during pregnancy was reported by mothers for three trimesters of pregnancy and was coded as non-smoking if the mother did not smoke during the entire pregnancy and smoking if the mother smoked at least during one trimester [74]. The mode of conception was classified in three groups: naturally conceived, conceived through stimulation, and conceived via in vitro fertilization (IVF) or intracytoplasmic sperm injection (ICSI) [75]. The mode of delivery was assessed as vaginal delivery, caesarean section (planned and urgent), and vaginal delivery with urgent intervention with forceps or vacuum extraction. Apgar scores at the 1st and 5th minute were presented in 3 conventional categories 0–6 (low), 7–9 (intermediate), and 10 (high) [76]. Gestational age, birthweight, parental age at birth were categorized in groups for descriptive statistics and treated as continuous z-scores in the analyses.

#### ALSPAC Replication Study

Breastfeeding was assessed via a questionnaire sent to mothers when the study children were approximately 6 months old. Breastfeeding was coded as ‘ever’ or ‘never’. Information on sample characteristics and covariates was obtained by questionnaires completed by the mother during pregnancy and from medical birth records. Socio-economic status was determined by the highest level of maternal education (grouped as follows: certificate of secondary education or not, vocational, O-level, A-level, or university degree). Maternal characteristics included age at birth, pre-pregnancy height and weight, and smoking during pregnancy (any or none). Gestational characteristics included caesarean delivery, gestational age and birthweight.

### 2.4. DNA Sample Collection

#### 2.4.1. NTR (Discovery and Replication Study)

DNA samples were collected from buccal swabs, as described previously [77]. In short, 16 cotton mouth swabs were individually rubbed against the inside of the cheek on 2 days (morning and evening) and placed in four separate 15 mL conical tubes containing 0.5 mL of a Sodium Chloride-Tris-Ethylenediaminetetraacetic acid (STE) buffer (100 mM sodium chloride, 10 mM Tris hydrochloride (pH 8.0), and 10 mM ethylenediaminetetraacetic acid) with proteinase K (0.1 mg/mL) and sodium dodecyl sulfate (SDS) (0.5%) per swab. Individuals were asked to refrain from eating or drinking 1 h prior to sampling. High molecular weight genomic DNA was extracted by standard DNA extraction techniques. The DNA samples were quantified using the Quant-iT PicoGreen dsDNA Assay Kit (ThermoFisher Scientific, Waltham, MA, USA).

#### 2.4.2. ALSPAC Replication Study

The ALSPAC blood-based DNA methylation profiles were generated at age 7 as part of the ARIES [71], a subsample of approximately 1000 mother-child pairs from the ALSPAC study.

### 2.5. DNA-Methylation Measurements

#### 2.5.1. NTR Discovery Study

The genome-wide DNA methylation in buccal cells [68] was assessed with the Infinium MethylationEPIC BeadChip (Illumina, San Diego, CA, USA) [78] by the Human Genotyping facility (HugeF) of ErasmusMC, the Netherlands (http://www.glimdna.org/). Five hundred nanograms of genomic DNA from buccal swabs were bisulfite treated using the ZymoResearch EZ DNA Methylation kit (Zymo Research Corp, Irvine, CA, USA). The quality control of the methylation data are described in detail elsewhere [68]. In brief, the quality control (QC) and normalization of the methylation data were performed using a pipeline developed by the Biobank-based Integrative Omics Study (BIOS) consortium [79], which includes sample quality control using the R package MethylAid [80] and probe filtering and functional normalization as implemented in the R package DNAmArray. We previously and successfully applied this pipeline in a pilot study of EPIC array data from buccal samples [63], which was used as replication sample in the current paper. MethylAid was applied with the default EPIC array-specific quality filter thresholds for EPIC arrays. The R package omicsPrint [81] was used to call genotypes based on methylation probes and to verify sample relationships based on those single nucleotide polymorphisms (SNPs) (e.g., the zygosity of twins and samples from the same individual). We checked for sample mismatches between methylation data and genotype data by computing the correlation between SNP genotypes called by omicsPrint based on methylation probes and genotypes based on genome-wide SNP arrays. DNAmArray and meffil [82] were used to identify sex mismatches (both packages identified the same mismatches).

Functional normalization was performed based on five control probe principal components (PCs). The following probe filters were applied: Probes were set to missing (NA) in a sample if they had an intensity value of exactly zero, detection *p*-value > 0.01, or bead count <3. Probes were excluded from all samples if they mapped to multiple locations in the genome, if they overlapped with an SNP or insertion/deletion (INDEL), or if they had a success rate <0.95 across samples. Annotations of ambiguous mapping probes (based on an overlap of at least 47 bases per probe) and probes where genetic variants (SNPs or INDELS) with a minor allele frequency >0.01 in Europeans overlap with the targeted CpG or single base extension site (SBE) were obtained from Pidsley et al. [83]. After probe filtering, the success rate of probes for each sample was checked: All samples had a success rate above 0.95 after removal of low-performing samples detected by MethylAid. Only the autosomal methylation sites were analyzed, leaving 787,711 out of 865,859 sites for analysis.

#### 2.5.2. NTR Replication Study

Genome-wide DNA methylation in buccal cells was assessed with the Infinium MethylationEPIC BeadChip (Illumina, San Diego, CA, USA) [78] by the Avera Institute for Human Genetics. The quality control, processing and normalization of the data were performed with the same pipeline as for the NTR discovery cohort, a pipeline which has previously been described in detail [63].

#### 2.5.3. ALSPAC Replication Study

DNA methylation wet laboratory procedures, preprocessing analyses, and quality control were performed at the University of Bristol, as previously described [71]. DNA methylation outliers were identified as those three times the inter-quartile range from the nearest of the first and third quartiles. Outliers were replaced with missing values.

### 2.6. Cellular Proportions

#### 2.6.1. NTR Discovery and Replication Study

Cellular proportions were predicted with hierarchical epigenetic dissection of intra-sample-heterogeneity (HepiDISH) with the RPC method (reduced partial correlation), as described by Zheng et al. [84] and implemented in the R package HepiDISH. HepiDISH is a cell-type deconvolution algorithm that was specifically developed for estimating cellular proportions in epithelial tissues based on genome-wide methylation profiles and that makes use of reference DNA methylation data from epithelial cells, fibroblast and seven leukocyte subtypes. This was applied to the data after data QC and normalization.

#### 2.6.2. ALSPAC Replication Study

Cell count estimates were estimated from DNA methylation profiles using a deconvolution algorithm [85] and included in statistical models to adjust for cell count heterogeneity.

### 2.7. Data Analyses

#### 2.7.1. Associations between Breastfeeding and Pre- and Perinatal Factors

The association between covariates and breastfeeding was tested in the discovery cohort using a generalized estimating equations (GEE) model that accounted for the correlation structure within families. Six breastfeeding duration categories were included as the continuous outcome variable. The predictors consisted of variables previously described as covariates in EWASs of maternal smoking and birth weight [56,57], i.e., parental SES, maternal smoking during pregnancy (yes/no), gestational age (*z*-scores), maternal age at birth (*z*-score), maternal pre-pregnancy BMI (*z*-scores), cell counts of epithelial cells and natural killer (NK) cells, child’s sex, and child’s age at DNA-methylation (see Appendix A). This analysis was performed in SPSS version 25.

#### 2.7.2. EWAS

##### Discovery Study

The association between DNA methylation level and breastfeeding was tested using a generalized estimating equations (GEE) model that accounted for the correlation structure within families with DNA methylation β-value as the outcome variable (see Appendix A). All analyses in the NTR were performed with GEE models, which were fitted with the R package ‘gee.’ The following settings were used: Gaussian link function (for continuous outcome variables), 100 iterations, and the ‘exchangeable’ option to account for the correlation structure within families. To examine and adjust (where applicable) for the inflation of test statistics, the R package bacon was used [86]. We first fitted a basic model with the following predictors: breastfeeding, sex, age at sample collection, the percentages of epithelial and natural killer cells, EPIC array row and bisulfite sample plate (using dummy coding). For the primary EWAS, the breastfeeding outcome was dichotomized (yes/no). Secondly, we fitted a model in which we adjusted for additional covariates (SES, maternal smoking, mother’s age at birth, mother’s BMI at pregnancy, and gestational age). These covariates were selected on the base of adjustments done in recent meta-analyses of maternal smoking and birthweight [56,57]. We also considered the effects of the duration of breastfeeding by evaluating the same model with the original six categories (not-breastfed; <2 weeks; 2–6 weeks; 6 weeks to 3 months; >6 months). Thirdly, we repeated the EWAS in children younger than 10 years and older than 10 years. To this end, we split up the sample into two groups (median age split) to have two equally-sized groups: children younger than 10 years and children older than 10 years. Epigenome-wide significance was assessed following Bonferroni correction for the number of methylation sites tested (α = 0.05/787,711 = 6.34 × 10^–8^).

We carefully checked if epigenome-wide significant associations were influenced by outliers. As a sensitivity analysis, we repeated the association analysis in the discovery sample for significant CpGs without outliers defined using the Tukey method [87], in which an outlier is any value greater than the upper quartile plus three-times the interquartile range or any value lower than the lower quartile minus three-times the interquartile ranges.

##### Twin Correlations

The correlation between DNA methylation levels of monozygotic (MZ) and dizygotic (DZ) twins was computed for CpGs that were significant in the discovery study and which were robust to outliers. The methylation beta-values were adjusted for the set of covariates as included in the basic EWAS model with a linear model, and residuals were saved. Next, the MZ and DZ twin correlations were computed on the residuals.

##### Replication

CpGs that were significant in the discovery study and which were robust to outliers were selected for replication. The basic and adjusted models were applied as in the discovery sample: in the NTR, the GEE model had the same covariates; in the ALSPAC, cell count estimates for peripheral blood were included instead of the epithelium cells count; the sex, age of the child at blood collection around age 7 and the same covariates from the adjusted Model 2 were included; no siblings were included. In the ALSPAC, DNA methylation variation due to technical artefacts or unknown confounders were handled by including 20 surrogate variables generated from the DNA methylation data using the ‘sva’ R package [88], and associations were tested with linear models implemented by the limma package [89].

### 2.8. Methylation Data Annotation

To examine previously reported associations for epigenome-wide significant CpGs associated with breastfeeding, we looked them up in the EWAS atlas (https://bigd.big.ac.cn/ewas/tools; accessed on 3 October 2019 [90]) and the EWAS catalog (http://www.ewascatalog.org; accessed on 3 October 2019). To analyze the possible function of CpGs, we searched for overlaps with mQTLs in a study that analyzed EPIC array data from 102 buccal samples [63] to identify the associated SNPs, and we looked up these SNPs and genes in the genome-wide association study (GWAS) catalog (https://www.ebi.ac.uk/gwas/; accessed on 4 October 2019).

### 2.9. Overlap with Previous Findings

The follow-up analyses were done for CpGs that were previously associated with breastfeeding in EWASs or candidate gene studies. These included all CpGs on the EPIC array that were annotated to the genes *LEP* (*n*_CpGs_ = 23) [52,53,54,55,58], *IL4R* (*n*_CpGs_ = 37) [58], *CDKN2A* (*n*_CpGs_ = 32) [59], 1 CpG from the study of Hartwig et al. [60], and 4297 suggestive CpGs from the study of Naumova et al. [53]. In total, this resulted in a list of 4370 CpGs, of which 3859 CpGs were present (after QC) in our data. For these CpGs, we performed a look-up analysis in the total sample (*n* = 1006) and in the subsample younger than 10 years (*n* = 517) of the discovery NTR cohort. Significance was assessed after Bonferroni correction for the number of CpGs tested (α = 0.05/3858 = 1.29 × 10^–5^).

## 3. Results

### 3.1. Descriptive Statistics

Descriptive statistics are presented in Table 1. The total discovery sample consisted of 1006 children (mean age 9.5, SD = 1.89); 73.7% of the children were breastfed with different duration, and 26.3% never received breastfeeding. Most of the children were breastfed at least one month: 189 (25.5%) were breastfed for 2 weeks to 1.5 months, 181 (24.4%) were breastfed for 1.5–3 months, 148 (20%) were breastfed for 2–6 months, and 148 (20%) were breastfed more than 6 months. The subsample of children younger than 10 years old included 517 twins (mean age 7.9, SD = 1.1) (see Appendix A), and the group older than 10 years included 489 twins (mean age = 11.16, SD = 0.72) (see Appendix A). The majority of twin pairs was concordant for breastfeeding (99.4%). Among the discordant twin pairs were three pairs, of which one twin was not breastfed while their co-twin received breastfeeding and three pairs of which the co-twins experienced different durations of breastfeeding.

Breastfeeding duration (six categories) was significantly positively associated with socio-economic status (β = 0.353, standard error (SE) = 0.09, and *p* = 0.0004) and inversely associated with maternal smoking (β = −0.733, SE = 0.32, and *p* = 0.02) and gestational age (β = −0.194, SE = 0.09, and *p* = 0.03) (see Appendix A). Breastfeeding was not significantly correlated with maternal pre-pregnancy BMI (β = −0.117, SE = 0.75, and *p* = 0.119), maternal age at delivery (β = −0.016, SE = 0.09, and *p* = 0.86), and the cell composition of buccal swabs (β = 1.27, SE = 1.2, and *p* = 0.25 for count of epithelial cells; and β = 14.78, SE = 9.8, and *p* = 0.13 for count of natural killer cells).

### 3.2. Association Analysis Findings

First, genome-wide DNA-methylation analyses were performed to test the association between breastfeeding (never/ever) and DNA-methylation level while adjusting for sex, age at DNA sample collection, estimated proportion of epithelial cells, the estimated proportion of natural killer cells, the row of the sample on the chip, and the bisulfite plate (model 1). No epigenome-wide significant sites were identified (see Appendix A). Genome-wide EWAS test statistics showed no inflation (see Appendix A). DNA methylation was also not associated with the duration of breastfeeding (see Appendix A).

In the EWAS in children younger than 10 years with the same basic model, we identified four epigenome-wide significant CpGs (see Appendix A): cg25178826 (β = −0.026, SE = 0.003, and *p* = 8.04 × 10^–12^) is located in the 5’ untranslated region (5′UTR) region of *PRLR* (prolactin receptor) gene, cg12087956 (β = −0.03, SE = 0.005, and *p* = 1.8 × 10^–8^) is in the gene body of *CDAN1* (codanin 1), cg24192772 (β = −0.024, SE = 0.004, and *p* = 2.5 × 10^–8^) is in the gene body of *FOXK2* (forkhead box K2), and cg10142656 (β = −0.02, SE = 0.004, and *p* = 6.28 ×10^–8^) is mapped to the transcription start site (TSS1500) of *TRMT10B* (tRNA methyltransferase 10B). These four CpGs were not strongly associated with breastfeeding duration (see Appendix A). The plots of the methylation values for these four CpGs revealed several extreme methylation values in the data (see Appendix A and Appendix A), and the association with breastfeeding disappeared after outlier removal: cg25178826 β = −0.00006, SE = 0.001, and *p* = 0.98; cg12087956 β = 0.00004, SE = 0.0009, and *p* = 0.5; cg24192772 β = 0.0009, SE = 0.001, and *p* = 0.15; and cg10142656 β = 0.0001, SE = 0.0008, and *p* = 0.13 (see Appendix A).

Next, we performed EWASs of breastfeeding never/ever with adjustments for SES, maternal age at delivery, maternal pre-pregnancy BMI, maternal prenatal smoking, and gestational age (Model 2). One CpG, cg22491379, was significant in the total sample (β = −0.007, SE = 0.001, and *p* = 1.3 × 10^–9^) (see Figure 2a, Appendix A, Table 2 and Appendix A), and seven CpGs were significant in the children younger than 10 years (see Table 2 and Appendix A). The Manhattan plot for the EWAS in children younger than 10 years showed a peak on chromosome 21, which contains a cluster of CpGs just below the epigenome-wide significance threshold (see Figure 2d). Results from the same analysis for breastfeeding duration are shown in Appendix A.

Genome-wide EWAS test statistics from the analyses of Model 2 (in the total sample and in the children younger than 10 years) showed no inflation (see Figure 2a,b). For four out of the eight significant CpGs, the association disappeared after removal of outliers (see Appendix A). The four CpGs that were unaffected by outliers were selected for a replication analysis (see Appendix A): cg16279140 (β = −0.4, SE = 0.05, and *p* = 3.5 × 10^–15^), cg05823759 (β = 0.2, SE = 0.032, and *p* = 2.35 × 10^–10^), cg27284194(β = 0.64, SE = 0.12, and *p* = 2.9 × 10^–9^), and cg03995300 (β = 0.23, SE = 0.04, and *p* = 1.2 × 10^–8^) (see Appendix A). All four CpGs were significant in children younger than 10 years.

We next examined the association with breastfeeding in children older than 10 (Model 2). In this analysis, the genome-wide test statistics showed mild inflation (lambda = 1.11). After adjusting for inflation, four CpGs remained epigenome-wide significant, but none were significant after outlier removal. The four significant CpGs in the children younger than 10 years had smaller effects, and three CpGs showed inverse directions of effect in children older than 10: cg16279140 (β = 0.04, SE = 0.02, and *p* = 0.04), cg05823759 (β = −0.002, SE = 0.02, and *p* = 0.92), cg27284194 (β = 0.18, SE = 0.06, and *p* = 0.002), and cg03995300 (β = −0.003, SE = 0.02, and *p* = 0.89) (see Appendix A). The quantile-quantile plots (see Figure 2b,c) suggested a stronger association signal for breastfeeding in children younger than 10 years than in the older group. We computed the correlations between the regression coefficients (i.e., the methylation difference associated with breastfeeding) for the top 100 CpGs from each age group. These correlations were weak (*r* = 0.25 and *p* = 0.10 for the top 100 CpG from the EWAS in children <10 years and from the EWAS in children >10 years). This suggests that methylation profiles associated with breastfeeding are different in children younger than 10 years and older than 10 years.

We computed the twin correlations in the MZ and DZ pairs for the significant CpGs, as we assumed that equal exposure to breastfeeding should cause similarity in methylation level of twins. The correlations of methylation levels of all four sites were very high in MZ twins and almost twice as high as in DZ twins: cg16279140 rMZ = 0.906, rDZ = 0.505; cg05823759 rMZ = 0.953, rDZ = 0.609; cg27284194 rMZ = 0.972, rDZ = 0.462; and cg03995300 rMZ = 0.902, rDZ = 0.453. This pattern suggests that there are heritable influences on DNA methylation at these CpGs, as previously shown by other studies of DNA methylation heritability.

### 3.3. Replication Analysis of Our Findings in Other Samples

Both replication datasets were comparable to the discovery subsample of children younger than 10 years old (see Appendix A). As expected, the singleton children (ALSPAC) had a higher gestational age (~39 weeks vs. ~35.7), a higher birthweight (~3500 g vs. ~2400 g) and a lower proportion of caesarean deliveries (~9% vs. ~36.8%). We performed a replication analysis for significant CpGs unaffected by outliers from the adjusted Model 2 of the discovery analysis with the same set of covariates (see Table 3).

In the NTR replication cohort, for one CpG, the direction of effect changed (cg03995300, *ZNF232*, β = −0.050, SE = 0.033, and *p* = 0.13), and three other CpGs had the same direction of effect but were not significant: cg16279140 (β = −0.03, SE = 0.04, and *p* = 0.43), cg05823759 (β = 0.03, SE = 0.033, and *p* = 0.32), and cg27284194 (β = 0.07, SE = 0.054, and *p* = 0.21) (see Table 3, Appendix A).

In the ALSPAC, the most important difference to note were the different tissues used for DNA methylation profiling (peripheral blood) and the platform (the Illumina Infinium HumanMethylation450 Beadchip [Illumina, San Diego, CA, USA]). Because of the platform difference, roughly half of the CpG sites included in the discovery DNA methylation profiles were measured in the ALSPAC. In the adjusted Model 2, two of the four most strongly associated CpG sites were not included on the ALSPAC platform (cg16279140 and cg05823759), nor were any other sites within 1000 base pairs (bp) of these sites. The other two sites, cg27284194 and cg03995300, were included, but neither association was replicated (*p* > 0.19 and *p* > 0.77, respectively) in the adjusted model (see Table 3). An overall lack of replication was confirmed by low correlations between effect estimates in the discovery and the ALSPAC. In particular, we identified the 100 most strongly associated CpG sites in the discovery that were also present in the ALSPAC DNA methylation profiles. The correlation of effect estimates for these sites between studies was low (Pearson’s *r* = 0.13 with *p* = 0.2 for the adjusted model).

### 3.4. Replication Analysis of Findings from Previous EWAS

Of the 3858 CpGs from previous literature, four CpGs were associated with breastfeeding in our sample (α = 1.29 × 10^–5^, see Table 4). One site was significant in the total sample (see Appendix A): cg16387046 is located on chromosome 12 in the *MUCL1* (mucin like 1) gene (β = 0.027, SE = 0.005, and *p* = 4.9 × 10^–7^). Three sites were significant in the children younger than 10 years (see Appendix A): cg16704958 (β = 0.009, SE = 0.002, and *p* = 8.03 × 10^–6^) and cg11287055 (β = 0.056, SE = 0.01, and *p* = 4.9 × 10^–6^), located on chromosome 21 in the *VPS26C* (endosomal protein sorting factor C; previous name *DSCR3*) gene, and cg26479305 (β = 0.338, SE = 0.07, and *p* = 1.11 × 10^–5^), located on chromosome 12 in the *ATG101* (autophagy related 101, previous name *C12orf44*) gene. All these CpGs were previously reported in the association study by Naumova et al. [53] that was carried out in infants around two years old (mean age 25.7 months) with DNA methylation profiling in peripheral blood with Illumina EPIC BeadChip (Illumina, San Diego, CA, USA). The direction of association was positive for all four CpGs in both studies. CpGs located in/nearby *LEP* [52,55], *IL4R* [58] and *CDKN2A* [59], previously discussed as candidate genes for association with breastfeeding, were not significant in our study.

### 3.5. Methylation Data Annotation

For the significant CpGs from discovery and follow-up studies, we looked up the associations with nearby genetic variants (mQTLs) in the results from a previously published mQTL study of buccal-derived DNA in monozygotic twins [63]. Four CpGs were associated with mQTLs (cg27284194, associated with SNPs on chromosome 4: 927973–1039876; cg16279140, associated with SNPs on chromosome 14: 103823243–104186876; cg16387046, associated with SNPs on chromosome 12: 54830518–54896008; and cg26479305, associated with SNPs on chromosome 12: 52492131). Additionally, no significant mQTLs were found for four CpGs (cg03995300, cg05823759, cg11287055, cg16704958).

Next, we compared our results against all previously associated traits in EWASs (the EWAS atlas and the EWAS catalog) and GWASs (the GWAS catalog). One CpG (cg03995300) is mapped to *ZNF232* and has been previously associated with prenatal maternal smoking, sex, and ancestry [56,91,92]. The *ZNF232* locus was associated with a family history of Alzheimer’s disease in a GWA meta-analysis [93]. The intergenic CpG on chromosome 4 (cg27284194) has been previously associated with infertility [94,95]. In GWASs, the region 4: 927973–1039876 (harboring mQTLs for cg27284194) has been associated with a number of traits and diseases, including bone mineral density [96,97,98,99,100] and several metabolomic characteristics such as blood protein [101,102] and triglyceride [103] levels (See Appendix A). The intergenic CpG on chromosome 14 (cg16279140) did not appear in previous EWASs. In GWASs, SNPs on chromosome 14: 103823243–104186876 have been linked to amino acids levels as biomarkers of metabolic disorders [104] and other blood metabolites [105], and some disease and addictions including multisite chronic pain [106], metabolite changes in chronic kidney disease [107], bipolar disorder [108], alcohol consumption [109] and risk-taking behavior [110] (See Appendix A). The intergenic CpG on chromosome 7, cg05823759, has not been previously identified in an EWAS and is not associated with mQTLs.

Four CpGs that were replicated from previous studies on breastfeeding exposure have been reported in association with other traits and diseases. The *ATG10 (C12orf44)* gene, where cg26479305 is located, has been related to the autophagy pathway and cellular senescence in an EWAS [111]. It is related to regulators of inflammation (circulating cytokines and growth factors) [112], hematological traits in GWASs [113], and prenatal arsenic exposure [114].

CpGs (cg11287055, cg16704958) mapped to the *VPS26C (DSCR3)* have not been reported in other EWASs. *VPS26C* is a component of the retriever complex, which plays a role in cell surface processes such as cell migration, cell adhesion, nutrient supply and cell signaling [115].

## 4. Discussion

We aimed to study DNA methylation in buccal cells in mid-childhood in association with breastfeeding as an exposure. Associations were tested in the total sample and in groups stratified by age. The age distribution allowed us to analyze whether effects of breastfeeding can be attenuated with age, as this has been observed for other exposures in peripheral blood, including maternal smoking [56], birthweight [57] and breastfeeding [55].

We did not find evidence for robust epigenome-wide significant associations in the total sample (1006 children, age 5–12), but we did observe associations in the younger age group (before 10 years) that did not appear in the group of children of 10–12 years old. This suggests that epigenetic alterations in certain genomic regions might be associated with nutritional differences in the early postnatal life that tend to fade across childhood. However, since the CpGs were not replicated in the small, buccal-cells replication NTR dataset and in the large blood-cells replication dataset from the ALSPAC, further studies are required to follow up on these findings. It also remains to be examined if the associations reflect a causal effect of breastfeeding on methylation and whether these methylation differences might influence developmental trajectories and later-life health of the child.

The observation that associations between breastfeeding and methylation at some CpGs are age-dependent may be explained by age-related changes in DNA methylation that have been reported in several studies [116,117], including children in the age range of our study [118,119,120,121,122]. Similarly, Sherwood et al. [55] observed that breastfeeding is associated with the DNA methylation of the *LEP* gene at age 10 but not at age 18. Previous studies have identified DNA methylation signatures of prenatal nutrition in peripheral tissues at various life stages, even in adulthood. DNA methylation signatures in many genomic regions have been identified in middle-aged individuals who were exposed to the 1944/45 Dutch hunger winter at the time of conception, and many of these sites have been found to be related to growth, developmental processes and metabolism [18,19]. It is unclear if the effects of early life postnatal exposures can have similar long-term effects to prenatal exposures in humans, as most studies of early life influences on the epigenome have focused on prenatal exposures or have not examined long-term effects. None of the genes reported in studies of prenatal malnutrition (*SMAD7*, *CDH23*, *INSR*, *RFTN1*, *CPT1A*, *KLF13* [19], *IGF2* [18], and *LEP* [123]) were present among the top CpGs associated with breastfeeding in our study. Several explanations can be proposed. First, postnatal nutrition and prenatal nutrition exposures might differently influence the DNA methylation. Second, DNA methylation signatures induced by early nutrition exposure can be different across examined tissues (buccal cells in our study versus peripheral blood in previous famine studies). Third, the prenatal famine study examined the effects of an extreme exposure during the prenatal period.

The presence of associations with breastfeeding initiation (breastfeeding never vs. ever), and the absence of findings with breastfeeding duration suggests that the association depends on the exposure to breast milk rather than its duration. This could potentially be in line with previous findings that have shown that effects of exposure on DNA methylation occur only when the individual is exposed in certain sensitive life periods [118]. Our finding of stronger epigenetic associations with breastfeeding ever vs. never than with the duration of breastfeeding is also in line with studies providing evidence that any breastfeeding has stronger biological effects than the duration of breastfeeding, e.g., the impact of first maternal milk (colostrum) on immunoglobulins and further neonatal health, especially for small for gestational age and low-birthweight infants [22,24].

We observed associations at a total of eight CpG sites: cg16279140, chromosome 14, intergenic; cg05823759, chr7, intergenic; cg27284194, chromosome 4, intergenic; cg03995300, chromosome 17, *ZNF232*; cg16387046, chromosome 12, *MUCL1*; cg11287055, chromosome 21, *VPS26C* (*DSCR3*); cg16704958, chromosome 21, *VPS26C* (*DSCR3*); and cg26479305, chromosome 12, *ATG10* (*C12orf44*). Methylation levels at these CpGs are potentially influenced by both environmental influences, such as breastfeeding, and by genetic variation, although an alternative interpretation could be that the genomic regions of these CpGs affect breastfeeding function. Unfortunately, no GWAS on breastfeeding is available to verify this suggestion. It remains to be examined whether methylation differences induced by breastfeeding have an effect on the aforementioned traits. Interestingly, associations with breastfeeding have been previously reported in epidemiological studies for bone mineral content and bone mineral density [124] and metabolite profiles [125]; outcomes that have been associated with SNPs in the regions detected in our study as differentially methylated between breastfed and non-breastfed children. It remains to be examined whether methylation differences induced by breastfeeding have an effect on these traits.

Since the role of nutrition is systemic, it has been assumed that biomarkers must be present in different tissues. To study the epigenetic mechanisms of breastfeeding, previous studies have examined DNA from peripheral blood [52,53,54,55,58] and also from tumor tissue [59]. Buccal epithelium is of interest because it offers a non-invasive way of biosample collection for an epigenetic analysis. A number of studies have demonstrated the potential of buccal cells to study DNA methylation [126]. We previously showed that, although there is some correlation between DNA methylation in buccal and blood cells (*n* = 22, age = 18 years), it is low for most CpGs interrogated by the Illumina 450k array [127]. The methylation levels of two of the CpG sites that were included in the ALSPAC replication study were highly correlated in buccal and blood cells: cg27284194 (*r* = 0.864) and cg03995300 (*r* = 0.782). In spite of this, the associations with breastfeeding observed in the NTR were not replicated in the ALSPAC. We observed some overlap with the study of Naumova et al. [53] in peripheral blood that used the EPIC array, and these sites had medium correlations between DNA methylation in buccal and blood cells (cg16387046 *r* = 0.690, cg11287055 *r* = 0.413, and cg16704958 *r* = 0.328).

We observed that breastfeeding-associated CpGs show strong correlations in MZ twins and larger correlations in MZ twins compared to DZ twins. As the large majority of twin pairs in our study were almost always concordant for breastfeeding, it is expected that twins of both types show resemblance for DNA methylation levels at these sites. The larger correlation in MZ compared to DZ twins in our study suggests that these sites are also subject to heritable influences. In line with this observation, some of these CpGs are associated with mQTLs. It should be noted that breastfeeding and lactation itself are heritable traits. In previous twin studies, the heritability of initiation of breastfeeding ranged from 49% [128] to 70% [129]. DNA-methylation profiles may be influenced by several early life factors. In our study, almost all early environmental factors of interest were shared by twins: SES, maternal age at birth, maternal pre-pregnancy BMI, and maternal smoking during pregnancy and gestational age.

Some CpGs that did not reach epigenome-wide significance can have potential for further epigenetic studies of breastfeeding. The CpG cg22491379, located in the *PTPN4* on chromosome 2, was discovered in total sample of children of 5–12 years old and, after outlier removal, had suggestive significance (*p* = 5.78 × 10^–3^). *PTPN4* was listed in suggestive regions associated with breast morphology (breast size) [130]. Some top CpGs in the discovery study (non-adjusted model) are associated with genes that are involved in growth and metabolism. The *PRLR* (prolactin receptor) gene, located on chromosome 5, is involved in prolactin receptor activity and the growth hormone receptor signaling pathway. Growth hormone binds to the prolactin receptor, this being the basis of induction of lactation by growth hormone [131]. The *FOXK2* gene, located on chromosome 17, is a member of forkhead box transcription factors and is involved in glucose metabolism, aerobic glycolysis and autophagy [132], thus playing a role in metabolic reprogramming towards aerobic glycolysis [133]. It is associated lean body mass [134] and has been found to be hypomethylated in CpG islands in obese patients’ adipose tissues [135]. We observed a cluster of associations on chromosome 21, each individual association just below epigenome-wide significance. The *VPS26C* (*DSCR3*) gene is located on chromosome 21 and, as mentioned earlier, contains two CpGs previously associated with breastfeeding in blood [53].

The strengths of our study include the use of the buccal epithelium methylome to investigate breastfeeding for the first time, sample size, and the availability of replication data, including data from another tissue (blood). Furthermore, our exposure, breastfeeding, was assessed very shortly after it occurred, reducing the risk of measurement error and recall bias. Both the discovery and replication samples have been thoroughly phenotyped and extensively studied. Our study also has limitations. First, the findings in the discovery study were not replicated in the cohort of children with buccal cell DNA methylation of the same age, possibly due to the small size of this cohort (98 monozygotic twins). Second, the discovery sample was included in a study of aggressive behavior, and not primarily examined for the purposes of research on breastfeeding. Third, we did not have longitudinal DNA methylation data to measure the stability of effects; however, we examined the possible attenuation of breastfeeding effects through an age-stratified analysis. Fourth, there are currently no datasets available on gene expression in buccal cells; thus, we could not examine relationships between DNA methylation and transcription. A strength of the current analysis is that it used the Illumina EPIC array, which has much greater coverage than the 450k array. Importantly, some of the top CpGs identified in the discovery EWAS were all novel EPIC probes. The replication analysis in the ALSPAC on DNA methylation data from peripheral blood, however, used the 450k array and therefore did not permit look-up of the same CpGs. The findings of this study require further replication in cohorts with buccal epithelium and other tissues to improve our understanding of breastfeeding-associated methylation changes in different tissues, and the possible utility as a biomarker of early life nutrition. Finally, a difficulty in studies of breastfeeding is that breast milk composition is unique in each mother. Lactation is influenced not only by genetic variation but also by many environmental factors such as the mother’s nutrition, lifestyle, level of stress, and attachment to the child [24,25,136,137,138,139]. Future epigenetic studies of breastfeeding could stratify the breastfeeding sample on the basis of criteria of breast milk composition. This might be more informative for predicting a child’s outcome but might also be a better indicator of a mother’s well-being. The value of our research is that it combined breastfeeding, prenatal characteristics, and methylation data. In the future, results from studies that will integrate epigenomic data with genomics, transcriptomics, and metabolomics may be used to develop prediction models of long-term outcomes in child development and health. Understanding of the mechanisms associated with breastfeeding will help to develop interventions to improve children’s health and reduce risk of chronic disease by supporting breastfeeding or optimizing infant nutrition when breastfeeding is not possible.

## 5. Conclusions

Our study provided a first indication that breastfeeding as an early life environmental factor may be associated with epigenetic variation in buccal cells in children. The findings point at new candidate loci influenced by breastfeeding. Future studies are needed to investigate if the DNA methylation signatures are caused by breastfeeding or by other unmeasured confounders (including a genetic predisposition to give or receive breastfeeding and other aspects of prenatal or postnatal diet), whether they are influenced by percentage of breastfed meals, exclusive breastfeeding duration, breastmilk composition, etc., and what age-related mechanisms drive changes in the association between breastfeeding and methylation.

## Figures and Tables

**Figure 1 nutrients-11-02804-f001:**
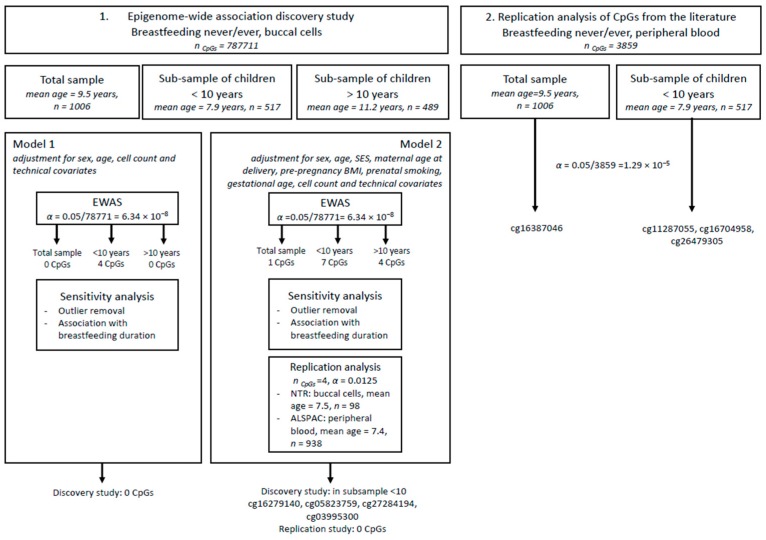
Flowchart of analyses.

**Figure 2 nutrients-11-02804-f002:**
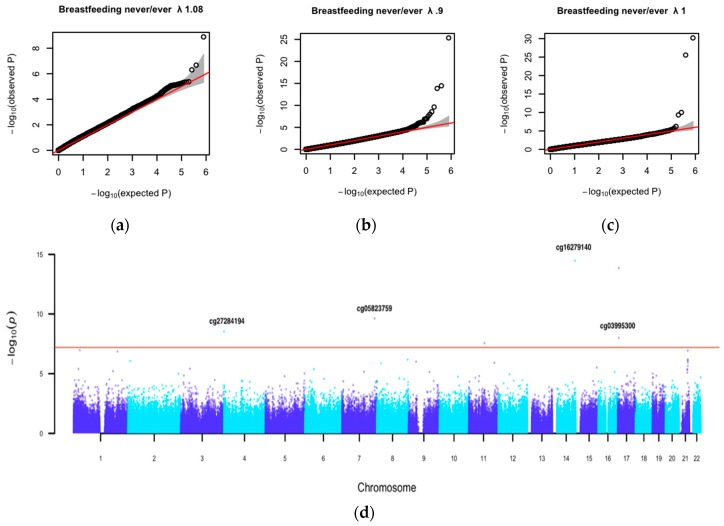
Quantile-Quantile (QQ) plots: (**a**) in the total sample; (**b**) in the subsample of children younger than 10 years old; and (**c**) in the subsample of children older than 10 years old. (**d**) Manhattan plot shows the epigenome-wide association study results of breastfeeding ‘never’/‘ever’ in children younger than 10 years old. Covariates included sex, age, socio-economic status (SES), maternal age at delivery, maternal pre-pregnancy BMI, maternal prenatal smoking, gestational age, EPIC array row and bisulfite sample plate, and cell composition. In the Manhattan plot, the red line represents the Bonferroni threshold (6.34 × 10^–8^). Given CpG names indicate significant loci after removal of outliers.

**Table 1 nutrients-11-02804-t001:** Early life characteristics and breastfeeding in the Netherlands Twin Register (NTR) discovery sample (*n* = 1006).

	Breastfeeding Never (*n* = 265)		Breastfeeding Ever (*n* = 741)		Total	
	*n*	%	*n*	%	*n*	%
Sex						
male	138	52.1%	349	47.1%	487	48.4%
female	127	47.9%	392	52.9%	519	51.6%
Zygosity						
Monozygotic (MZ)	226	85.3%	613	82.7%	839	83.4%
Dizygotic (DZ)	39	14.7%	128	17.3%	167	16.6%
Chorionicity						
MCMA	4	3.5%	21	7.3%	25	6.2%
MCDA	61	53.0%	144	50.3%	205	51.1%
DCDA	50	43.5%	121	42.3%	171	42.6%
Gestational Age (Weeks)						
Mean (SD)	36.2	(22.2)	35.8	(25.9)	35.9	(25.1)
≤ 32	12	4.8%	61	8.4%	73	7.5%
33–36	128	51.4%	359	49.4%	487	49.9%
≥ 37	109	43.8%	306	42.1%	415	42.6%
Mother’s Age at Birth (Years)						
Mean (SD)	31.9	(4.5)	31.2	(4.2)	31.4	(4.3)
19–29	76	29.0%	288	39.0%	364	36.4%
30–39	175	66.8%	435	58.9%	610	61.0%
>40	11	4.2%	15	2.0%	26	2.6%
Mother’s BMI Before Pregnancy						
Mean (SD)	24.3	(4.0)	24.2	(4.11)	24.2	(4.1)
<25	149	61.3%	470	66.1%	619	64.9%
25–29	70	28.8%	169	23.8%	239	25.1%
30–39	24	9.9%	65	9.1%	89	9.3%
>40	0	0.0%	7	1.0%	7	0.7%
Father’s Age at Birth (Years)						
Mean (SD)	33.2	(4.4)	33.9	(5.4)	33.7	(5.2)
20–29	53	22.0%	146	20.3%	199	20.8%
30–39	163	67.6%	482	67.1%	645	67.3%
>40	25	10.4%	90	12.5%	115	12.0%
Mode of Conception						
naturally	227	92.7%	623	86.5%	850	88.1%
stimulated	4	1.6%	26	3.6%	30	3.1%
IVF/ICSI	14	5.7%	71	9.9%	85	8.8%
Maternal Smoking						
no smoking	205	86.1%	631	92.9%	836	91.2%
smoking	33	13.9%	48	7.1%	81	8.8%
Parental SES						
low skill level	0	0.0%	8	1.2%	8	0.9%
lower secondary educational level	30	11.9%	41	6.1%	71	7.6%
upper secondary education level	99	39.3%	203	30.0%	302	32.5%
higher vocational level	89	35.3%	234	34.6%	323	34.8%
scientific level	34	13.5%	191	28.2%	225	24.2%
Mode of Delivery						
vaginal	141	56.6%	416	57.1%	557	57.0%
caesarean planned	43	17.3%	97	13.3%	140	14.3%
urgent intervention (forceps, vacuum extraction)	20	8.0%	75	10.3%	95	9.7%
urgent caesarean section	45	18.1%	140	19.2%	185	18.9%
Birth Weight						
Mean (SD)	2435.7	(444.8)	2394.6	(558)	2405	(531.7)
<1500	8	3.2%	52	7.1%	60	6.2%
1500–2500	123	49.8%	338	46.4%	461	47.3%
>2500	116	47.0%	338	46.4%	454	46.6%
Apgar Score at 1st Minute						
0–6	17	12.3%	48	12.8%	65	12.7%
7–9	103	74.6%	290	77.5%	393	76.8%
10	18	13.0%	36	9.6%	54	10.5%
Apgar Score at 5th Minute						
0–6	1	0.8%	14	3.9%	15	3.1%
7–9	41	31.1%	130	36.3%	171	34.9%
10	90	68.2%	214	59.8%	304	62.0%
Breastfeeding Duration						
no	265	100.0%	0		265	26.3%
less than 2 weeks			75	10.1%	75	7.5%
2–6 weeks			189	25.5%	189	18.8%
6 weeks to 3 months			181	24.4%	181	18.0%
3–6 months			148	20.0%	148	14.7%
more than 6 months			148	20.0%	148	14.7%

Descriptive statistics of children included in the study. MCMA = monochorionic monoamniotic; MCDA = monochorionic diamniotic; DCDA = dichorionic diamniotic; SD = standard deviation; BMI = body mass index; IVF = in vitro fertilization; ICSI = intracytoplasmic sperm injection; and SES = socio-economic status.

**Table 2 nutrients-11-02804-t002:** Summary of epigenome-wide significant CpGs from the discovery epigenome-wide association study (EWAS) of DNA signatures of breastfeeding.

cgID	Chromosome	Position	Gene	Gene Region	Discovery Study			Discovery Study Without Outliers		
					Estimate	SE	*p*-Value	Estimate	SE	*p*-Value
Basic Model (1). Sub-Sample < 10 years (*n* = 517)										
cg25178826	chr5	35165447	*PRLR*	5’UTR	−0.026	0.004	8.04 × 10^–12^	−6.04 × 10^–5^	0.001	0.98
cg12087956	chr15	43022167	*CDAN1*	Body	−0.031	0.005	1.18 × 10^–8^	4.92 × 10^–5^	0.001	0.50
cg24192772	chr17	80536920	*FOXK2*	Body	−0.024	0.004	2.52 × 10^–8^	9.17 × 10^–4^	0.001	0.15
cg10142656	chr9	37753047	*TRMT10B*	TSS1500	−0.019	0.004	6.28 × 10^–8^	9.98 × 10^–5^	0.0007	0.14
Adjusted Model (2). Discovery Sample (*n* = 1006)										
cg22491379	chr2	120553625	*PTPN4*	5’UTR	−0.007	0.001	1.30 × 10^–9^	−0.005	0.001	5.78 × 10^–^^3^
Adjusted Model (2). Sub-Sample < 10 Years (*n* = 517)										
cg03463465	chr6	164143581			0.360	0.034	4.51 × 10^–26^	−0.003	0.001	0.01
cg07670516	chr17	5019840	*ZNF232*	5’UTR	0.249	0.032	1.40 × 10^–14^	0.006	0.014	0.65
cg20820810	chr11	71850130	*FOLR3*	Body	−0.300	0.054	2.82 × 10^–8^	−0.001	0.001	0.21
cg16279140	chr14	103981749			−0.411	0.052	3.50 × 10^–15^	no outliers		
cg05823759	chr7	149646627			0.205	0.032	2.35 × 10^–10^	no outliers		
cg27284194	chr4	1044797			0.638	0.107	2.90 × 10^–9^	no outliers		
cg03995300	chr17	5019989	*ZNF232*	5’UTR	0.229	0.040	1.02 × 10^–8^	no outliers		

α = 0.05/787,711 = 6.34 × 10^–8^. Basic Model 1 included breastfeeding coded as ‘never’ and ‘ever’ with adjustments for sex, age at DNA methylation, the count of epithelial cells, the count of natural killer cells, EPIC array row and bisulfite sample plate. Adjusted Model (2) included in additional to basic model covariates: SES, maternal smoking during pregnancy, maternal age at birth, maternal pre-pregnancy BMI, and gestational age. In bold: CpGs selected for replication.

**Table 3 nutrients-11-02804-t003:** Summary of association between breastfeeding and significant in the discovery study CpGs in replication.

cgID	Direction of Effect in Discovery Study <10 Years	NTR Replication Study (*n* = 98)			ALSPAC Replication Study (*n* = 938)		
		Estimate	SE	*P*-Value ^a^	Estimate	SE	*P*-Value
cg16279140	−	−0.0326	0.0412	0.43	NA	NA	NA
cg05823759	+	0.0329	0.0332	0.32	NA	NA	NA
cg27284194	+	0.0668	0.0542	0.21	0.0047	0.016	0.77
cg03995300	+	−0.0502	0.0334	0.13	0.0140	0.011	0.19

Adjusted Model 2 was used. ^a^ α = 0.05/4 = 0.01. NA = not available on the 450k platform.

**Table 4 nutrients-11-02804-t004:** Replication of CpGs from previous literature.

cgID	Chromosome	Position	Gene	Gene Region	ESTIMATE	SE	*P*-Value
Total Sample (*n* = 1006)							
cg16387046	chr12	55248207	*MUCL1*	TSS200	0.027	0.005	4.93 × 10^–7^
Sub-Sample <10 (*n* = 517)							
cg11287055	chr21	38630234	*VPS26C (DSCR3)*	Body	0.056	0.012	4.93 × 10^–6^
cg16704958	chr21	38630728	*VPS26C (DSCR3)*	Body	0.009	0.002	8.03 × 10^–6^
cg26479305	chr12	52470979	*ATG10 (C12orf44)*	3’UTR	0.338	0.077	1.11 × 10^–5^

α = 0.05/3859 = 1.29 × 10^–5^. The table shows all CpGs that were previously reported to be associated with breastfeeding and that were significantly replicated by our study.

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
