# Peer review of "DNA Methylation Signatures of Breastfeeding in Buccal Cells Collected in Mid-Childhood"

_nutrients, 2019, doi:10.3390/nu11112804_

Round 1

Reviewer 1 Report

This is a well-written, interesting, and statistically sound project. I appreciated that analyses were performed in several cohorts and multiple tissue types.

I only offer a couple considerations for the discussion.

There is quite a bit of variability in breastfeeding literature as to how breastfeeding is defined: never vs. ever, duration of (any) breastfeeding, percentage of breastfed meals, and exclusive breastfeeding duration. Each have been associated with various health, cognitive, and social outcomes in infants. The authors may consider discussing this a bit further. Is there a particular reason they believe that "never" vs. "ever" breastfeeding might yield the strongest epigenetic differences, or might other breastfeeding variables yield significant, but perhaps different, findings? For example, recent work suggests that exclusive breastfeeding duration may have a particular impact on aspects of infant social development. 

Reviewer 2 Report

Large number of cohorts with successful genetic testing technique were used in this study to study the effect of breast feeding on epigenetics. The authors could not detect any significant loci in total samples, but found some clue in the samples of younger ages. This seems realistic.

1) In NTR breast feeding was for 2 years and in ALSPAC study breast feeding was for 6 months, can these difference be accepted as same feeding strategy?

2) Why the authors divided the subjects by 10 years of age? Any physiological development happened at time? If secondary sex characteristics considered, the male and female should be studied in the different cutoff.

3)Page3. line119: maybe typographical error. 

including breastfeeding 53–56, 
